# Association between socioeconomic status and self-reported, tested and diagnosed COVID-19 status during the first wave in the Northern Netherlands: a general population-based cohort from 49 474 adults

Yinjie Zhu [1], Ming-Jie Duan,[1] Hermien H. Dijk,[2] Roel D. Freriks,[2] Louise H. Dekker,[1,3] Jochen O. Mierau,[2,3] Lifelines Corona Research initiative[2,4,5,6]

For numbered affiliations see end of article.

**Correspondence to**
MSc Yinjie Zhu; y.zhu@umcg.nl

## ABSTRACT

**Objectives** Studies in clinical settings showed a potential relationship between socioeconomic status (SES) and lifestyle factors with COVID-19, but it is still unknown whether this holds in the general population. In this study, we investigated the associations of SES with self-reported, tested and diagnosed COVID-19 status in the general population.

**Design, setting, participants and outcome measures** Participants were 49 474 men and women (46±12 years) residing in the Northern Netherlands from the Lifelines cohort study. SES indicators and lifestyle factors (i.e., smoking status, physical activity, alcohol intake, diet quality, sleep time and TV watching time) were assessed by questionnaire from the Lifelines Biobank. Self-reported, tested and diagnosed COVID-19 status was obtained from the Lifelines COVID-19 questionnaire.

**Results** There were 4711 participants who self-reported having had a COVID-19 infection, 2883 participants tested for COVID-19, and 123 positive cases were diagnosed in this study population. After adjustment for age, sex, lifestyle factors, body mass index and ethnicity, we found that participants with low education or low income were less likely to self-report a COVID-19 infection (OR [95% CI]: low education 0.78 [0.71 to 0.86]; low income 0.86 [0.79 to 0.93]) and be tested for COVID-19 (OR [95% CI]: low education 0.58 [0.52 to 0.66]; low income 0.86 [0.78 to 0.95]) compared with high education or high income groups, respectively.

**Conclusion** Our findings suggest that the low SES group was the most vulnerable population to self-reported and tested COVID-19 status in the general population.

## Strengths and limitations of this study

► This study added evidence to the socioeconomically patterned COVID-19 status in a general population instead of in clinical settings.

► This study innovatively included a broader range of COVID-19 status, including self-reported and tested COVID-19 status, to better understand COVID-19-related socioeconomic factors.

► This study might fail to identify the ethnic differences because the study population was predominantly white (more than 98%).

► During the first wave of the COVID-19 pandemic in the Northern Netherlands, a number of cases identified were relatively low compared with the rest of the country, which might limit the power of the analysis about diagnosed COVID-19 status.

## INTRODUCTION

The ongoing COVID-19 pandemic has already infected 62 363 527 people worldwide (as of 1 December 2020).[1] Many studies have reported a relationship between lifestyle factors (e.g., smoking[2] and unhealthy diets[3]) or comorbidities associated with lifestyle factors (e.g., obesity[4] and diabetes mellitus[5]) and worse outcomes for COVID-19. Since lifestyle factors are often associated with socioeconomic status (SES),[6] several studies have revealed that people with low SES are more susceptible to COVID-19 infection, hospitalisation and mortality,[7–11] the interplay between SES, lifestyle and COVID-19 status becomes a valid query and it is likely that SES is a determinant of COVID-19.

Therefore, a rising concern among policy-makers and public health practitioners is that the COVID-19 pandemic might exacerbate the persistent socioeconomically patterned health inequalities over the life course.[12 13] Still, an in-depth investigation of the lifestyle and socioeconomic determinants of COVID-19 in the general population is needed because current COVID-19 studies mainly relied on individuals presenting with symptoms in a

clinical setting and focused on post-COVID-19 infection status (e.g., hospitalisation and intensive care).[14 15] As such, we do not have enough knowledge on the validity of these early results for the general population.

This study aimed to investigate the extent to which SES and lifestyle factors were associated with a broader range of COVID-19 status, that is, self-reported COVID-19, tested COVID-19 and diagnosed COVID-19 in the general population.

## METHODS

### Study design and participants

To assess COVID-19 status in the general population, we used data from the Lifelines COVID-19 Cohort. The Lifelines COVID-19 Cohort is a questionnaire-based study in a part of the Dutch Lifelines cohort.[16 17] To study the relationship between COVID-19 and SES, we linked data from the Lifelines COVID-19 questionnaire with data from the general Lifelines cohort, which contains a wide range of background data, such as demographics, lifestyles and SES.[18 19] For the present study, 49 474 participants from both the Lifelines cohort and the Lifelines COVID-19 Cohort, who had available and reliable data on demographics, SES and lifestyle, were included in the analysis (online supplemental figure 1).

The Lifelines cohort study is a multidisciplinary prospective population-based cohort study based on a unique three-generation design. The participants were recruited from the three Northern provinces of the Netherlands between 2006 and 2013 and the first group of participants were recruited via local general practitioners. The participants could indicate whether their family members were interested to be recruited as well. In addition, individuals who were interested in the study could also participant through online self-registration. Participants with insufficient knowledge of the Dutch language, with severe psychiatric or physical illness and those with limited life expectancy (<5 years) were excluded from the study. The Lifelines cohort study employs a broad range of investigative procedures in assessing the biomedical, sociodemographic, behavioural, physical and psychological factors that contribute to health and disease. A detailed description of the Lifelines cohort study can be found elsewhere.[18 19] Before study entry, a signed informed consent form was obtained from each participant. Adult participants (≥18 years) were asked to complete several questionnaires regarding various aspects, including demographics, SES and lifestyle.

The Lifelines COVID-19 Cohort collected data about COVID-19-related symptoms, current health issue and societal impacts from participants recruited from the Lifelines cohort. It is developed based on a COVID-19 questionnaire to identify genetic and environmental risk factors for COVID-19 and address the medical, social and psychological aspects of the pandemic. A detailed description of the Lifelines COVID-19 Cohort and the COVID-19 questionnaire can be found elsewhere.[16] The COVID-19

questionnaire is sent out weekly since 30 March 2020 for 12 weeks. Of the 139 713 Lifelines participants invited, 74 268 (53.2%) completed at least one of the questionnaires in the 12-week programme.

### COVID-19 status

There were three different COVID-19 statuses obtained in our study from the COVID-19 questionnaire: self-reported COVID-19, tested COVID-19 and diagnosed COVID-19. All COVID-19 statuses were coded as binary variables. Self-reported COVID-19 status was obtained by asking 'if you must choose, do you think you have (or have had) a COVID-19 infection?' or 'has a doctor told you that you probably have (had) a coronavirus/COVID-19 infection'; tested COVID-19 status was obtained from the question 'have you been tested for coronavirus (COVID-19)?'; diagnosed COVID-19 status was defined by asking 'do you have or have you had a coronavirus/COVID-19 infection?' or 'what was the results of your corona virus (COVID-19) test?' when they had a COVID-19 test.

Since some of the COVID-related questions were worded such that it is not possible to infer whether the answer referred to the period between the last questionnaire and the current questionnaire or to the beginning of the pandemic and the current questionnaire (e.g., "have you been tested for coronavirus (COVID-19)?"), the 12 questionnaires were condensed into a single observation per individual, indicating for each question whether the individual had at any point answered 'yes' during the study period.

### Socioeconomic status

SES was indicated by (a) highest educational level achieved and (b) monthly net household income level separately from the Lifelines cohort study at baseline. Highest educational level achieved was categorised as: (1) low—junior general secondary education or lower (International Standard Classification of Education [ISCED] level 0, 1 or 2); (2) middle—secondary vocational education and senior general secondary education (ISCED level 3 or 4) and (3) high—higher vocational education or university (ISCED level 5 or 6)[20]; household net income level was categorised as: (1) low: <2000 euro/month; (2) middle: 2000–3000 euro/month and (3) high: >3000 euro/month.

### Lifestyle factors

Six lifestyle factors (i.e., smoking status, alcohol consumption, diet quality, physical activity, TV watching time and sleep time) were selected from the Lifelines cohort study at baseline. Smoking status was categorised into never, former and current smoker. Alcohol intake and dietary consumption were derived from a validated 110-item semiquantitative food frequency questionnaire (FFQ) that assessed food consumption over the past month.[21] Heavy drinking was defined as >40 g or >20 g average per day alcohol consumption for men and women, respectively.[22] Lifelines Diet

Score was calculated to assess the overall diet quality based on the FFQ. This score ranks the relative intake of nine food groups with positive health effects (vegetables, fruit, whole grain products, legumes/nuts, fish, oils/soft margarines, unsweetened dairy, coffee and tea) and three food groups with negative health effects (red/processed meat, butter/hard margarines and sugar-sweetened beverages). The development of this score is described in detail elsewhere.[23] Nonoccupational moderate-to-vigorous physical activity was calculated in minutes per week from the validated Short Questionnaire to ASsess Health-enhancing physical activity data, which incorporated leisure time and commuting physical activities, including sports, at moderate (4.0–6.4 metabolic equivalent of task [MET]) to vigorous (≥6.5 MET) intensity.[24] TV watching time and sleep time were self-reported. Body mass index (BMI) was calculated by dividing weight in kilograms by the square of height in metres. The BMI was additionally categorised into underweight (BMI<18.5 $kg/m^2$), normal (18.5≤BMI<25 $kg/m^2$), overweight (25≤BMI<30 $kg/m^2$) and obese (BMI≥30 $kg/m^2$).[25]

## Statistical analysis

Nominal variables are presented as percentage (%). Continuous variables were shown as mean±standard deviation (SD) or median (interquartile range [IQR]). P values <0.05 were considered statistically significant. To analyse the associations of SES (i.e., education and income) with self-reported and tested COVID-19 status, we used logistic regression models to estimate ORs. Furthermore, we fitted robust Poisson regression models to estimate relative risk for associations between SES and diagnosed COVID-19 since the number of individuals with diagnosed COVID-19 was relatively low in this population. For all regression models, SES indicators were first entered into the model plus age and sex (model 1) and then adjusted for other covariates (model 2: model 1 plus six lifestyle factors; model 3: model 2 plus BMI; model 4: model 3 plus ethnicity). All statistical analyses were conducted using Stata, V.13.1 (StataCorp, Texas, USA) or RStudio, V.4.0 (RStudio PBC, Boston, USA). Sensitivity analyses were conducted by further adjusting for the total number of questionnaires filled by each participant (online supplemental table 1), by comparing the characteristics of study population and excluded participants (online supplemental table 2), and by treating lifestyle factors together as a mediator in the pathway between SES and COVID-19 status (online supplemental table 3).

## Patient and public involvement

Patients and/or the public were not involved in the design, or conduct, or reporting or dissemination plans of this research.

## RESULTS

Participant characteristics are shown in table 1. The average age of the study population was 46±12 years old and more women were recruited in this study (59.7%). More than 92% of the population characterised their ethnicity as White. The prevalence of overweight and obese was 40.0% and 14.5%, respectively (table 1). Out of the 49 474 participants who were recruited in both Lifelines cohort and Lifelines COVID-19 cohort, 4711 participants self-reported a COVID-19 infection, while 2833 participants reported having had a COVID-19 test and 123 participants reported having had a positive outcome of the COVID-19 test (table 1). Self-reported, tested and diagnosed COVID-19 participants were more likely to be female, to have attained middle or high education, to have high income and to have never smoked (table 1).

The association between education and different COVID-19 status is presented in table 2. Participants with low and middle education level were less likely to self-report COVID-19 (OR [95% CI]: low 0.78 [0.71 to 0.86]; middle 0.90 [0.84 to 0.97]) and be tested for COVID-19 (OR [95% CI]: low 0.58 [0.52 to 0.66]; middle 0.72 [0.66 to 0.79]) compared with highly educated participants after adjustment for age, sex, lifestyle factors, ethnicity and BMI (table 2). In addition, compared with high education participants, participants with middle education level had almost two-fold higher risk to be infected with COVID-19 (OR [95% CI]: 1.73 [1.19 to 2.50]) (table 2, model 1) once they had been tested for COVID-19. The association was slightly attenuated after adjustment for lifestyle factors (OR [95% CI]: 1.69 [1.17 to 2.44]) (table 2, model 2) and lifestyle factors seemed to slightly mediate the association between self-reported COVID-19 status (proportion mediated: 9.9%) (online supplemental table 3), however, further adjustment for BMI and ethnicity did not change the results.

The association between income and different COVID-19 status is presented in table 3. Low or middle income groups were less likely to self-report COVID-19 (OR [95% CI]: low 0.86 [0.79 to 0.93]; middle 0.80 [0.74 to 0.87]) and to be tested for COVID-19 (OR [95% CI]: low 0.86 [0.78 to 0.95]; middle 0.86 [0.78 to 0.94]) compared with high-income group after adjustment for all covariates (table 3, model 4). Moreover, lifestyle factors only slightly mediated the association between self-reported and tested COVID-19 statuses (proportion mediated: 6.7% and 9.5%, respectively) (online supplemental table 3). Nevertheless, no risk differences were found between different income levels and diagnosed COVID-19 status, once tested.

**Table 1** Demographics, socioeconomic status and lifestyle of the study population

| | COVID-19 status | | | |
| | Self-reported (n=4711) | Tested (2883) | Diagnosed (123) | Total response (n=49 474) |
|---|---|---|---|---|
| Sex, male% | 37.4 | 30.4 | 30.1 | 40.3 |
| Age, mean±SD | 42±11 | 44±12 | 44±9 | 46±12 |
| Ethnicity, white% | 89.4 | 92.8 | 94.3 | 92.0 |
| BMI, mean±SD | 25.9±4.3 | 25.7±4.3 | 25.7±3.8 | 26.0±4.2 |
| Underweight,% | 0.7 | 1.0 | 0.8 | 0.7 |
| Normal,% | 46.7 | 48.0 | 47.2 | 44.9 |
| Overweight,% | 37.2 | 36.6 | 40.7 | 40.0 |
| Obese,% | 15.4 | 14.5 | 11.4 | 14.5 |
| Education,% | | | | |
| Low | 18.0 | 16.4 | 14.6 | 23.5 |
| Middle | 40.8 | 37.4 | 48.0 | 39.2 |
| High | 41.2 | 46.2 | 37.4 | 37.3 |
| Income, % | | | | |
| Low | 30.1 | 28.3 | 23.6 | 28.1 |
| Middle | 29.5 | 30.7 | 33.3 | 33.7 |
| High | 40.4 | 41.0 | 43.1 | 38.2 |
| Smoking, % | | | | |
| Current | 20.5 | 15.1 | 15.5 | 16.9 |
| Former | 33.0 | 36.1 | 40.7 | 36.3 |
| Never | 46.5 | 48.9 | 43.9 | 46.8 |
| TV watching time ≥4 hours/day,% | 15.2 | 15.1 | 18.7 | 16.7 |
| Sleep time <7 or>9 hours/day, % | 16.3 | 14.4 | 13.8 | 14.9 |
| MVPA <150 min/week,% | 38.8 | 37.5 | 29.3 | 37.9 |
| LLDS | 24.3±6.1 | 25.1±6.0 | 25.4±5.9 | 24.7±6.0 |
| Alcohol, g/day | 4.9 (1.3–11.0) | 3.9 (0.9–10.0) | 3.5 (0.9–9.0) | 4.9 (1.2–11.1) |
| Heavy drinker, % | 3.1 | 2.8 | 2.4 | 2.6 |

BMI, body mass index; LLDS, lifelines diet score; MVPA, non-occupational moderate-to-vigorous physical activity; SD, standard deviation.

## DISCUSSION

This paper has provided evidence and insight into socioeconomic disparities in self-reported COVID-19, tested COVID-19 and diagnosed COVID-19 in a general population. We found that low and middle SES groups, especially participants with low education, were less likely to self-report COVID-19 and to be tested for COVID-19. Moreover, participants with middle education were more likely to be infected with COVID-19 compared with those who obtained high education.

Our study highlighted the importance of mitigating the socioeconomically patterned effect of the COVID-19 pandemic by showing a clear relationship between SES and COVID-19 status, which enables and aids policymakers to adequately respond to the pandemic and develop public health measurements. It is worrisome that low SES group is likely to be neglected or undetected during the pandemic because they are less likely to self-report and be tested for COVID-19. Given the fact that they are also at high risk of noncommunicable diseases,[26] once infected

with COVID-19 or other infectious diseases in the future, individuals with low SES might suffer disproportionately from the disease development compared with the high SES group, which may further exacerbate health inequality.[27] Therefore, national or regional commitment of public health measures, guidelines or interventions are needed to promote health equity before, during and after COVID-19 pandemic by identifying, targeting and engaging the low SES group.

One of the potential mechanisms of the inverse association between SES and COVID-19 status is poor health literacy. Health literacy refers to 'a person's knowledge, motivation and competences to access, understand, appraise and apply health information in order to make judgements and take decisions in everyday life'.[28] It is known that low SES is the most important determinant of health literacy.[29] During the COVID-19 pandemic, people with high SES may be more aware of the symptoms and health consequences of COVID-19, and can, therefore, better interpret and communicate and make a decision about their own health condition.[30 31] In other words, low SES people

**Table 2** Association of education with different COVID-19 status

| Education | COVID-19 status | | | | | |
|---|---|---|---|---|---|---|
| | Self-reported | | Tested | | Diagnosed | |
| | OR (95% CI) | p | OR (95% CI) | p | RR (95% CI) | p |
| Model 1 | | | | | | |
| Low | 0.81 [0.75–0.89] | <0.001 | 0.58 [0.52–0.65] | <0.001 | 1.18 [0.67–2.07] | 0.6 |
| Middle | 0.93 [0.87–0.99] | 0.03 | 0.73 [0.67–0.80] | <0.001 | 1.73 [1.19–2.50] | 0.004 |
| High | Ref | | | | | |
| Model 2 | | | | | | |
| Low | 0.79 [0.72–0.86] | <0.001 | 0.59 [0.53–0.66] | <0.001 | 1.11 [0.61–2.00] | 0.7 |
| Middle | 0.91 [0.85–0.97] | 0.006 | 0.74 [0.68–0.81] | <0.001 | 1.69 [1.17–2.44] | 0.005 |
| High | Ref | | | | | |
| Model 3 | | | | | | |
| Low | 0.77 [0.71–0.85] | <0.001 | 0.59 [0.53–0.66] | <0.001 | 1.11 [0.61–2.00] | 0.7 |
| Middle | 0.90 [0.84–0.96] | 0.002 | 0.74 [0.68–0.81] | <0.001 | 1.69 [1.16–2.45] | 0.006 |
| High | Ref | | | | | |
| Model 4 | | | | | | |
| Low | 0.78 [0.71–0.86] | <0.001 | 0.58 [0.52–0.66] | <0.001 | 1.10 [0.60–2.01] | 0.8 |
| Middle | 0.90 [0.84–0.97] | 0.004 | 0.72 [0.66–0.79] | <0.001 | 1.77 [1.21–2.60] | 0.003 |
| High | Ref | | | | | |

Model 1: adjusted for education, age and sex. Model 2: adjusted for education, age, sex, and six lifestyle factors (smoking status, TV watching time ≥ 4h/day, sleep time <7 or >9 h/day, MVPA<150 min/week, LLDS, and heavy drinker). Model3: adjusted for education, age, sex, six lifestyle factors (smoking status, TV watching time ≥ 4h/day, sleep time <7 or >9 h/day, MVPA<150 min/week, LLDS, and heavy drinker), and BMI. Model 4: adjusted for education, age, sex, six lifestyle factors (smoking status, TV watching time ≥ 4h/day, sleep time <7 or >9 h/day, MVPA<150 min/week, LLDS, and heavy drinker), BMI, and ethnicity.
BMI, body mass index; LLDS, lifelines diet score; MVPA, non-occupational moderate-to-vigorous physical activity; OR, odds ratio; RR, risk ratio.

might be less sensitive and aware of their COVID-19-related symptoms, or they might lack the ability to assess whether they could have had COVID-19 infection. Since they might lack health literacy, low SES people might be less likely to seek medical help even when they have symptoms because of the lack of practicing health-seeking behaviour and critical health resources.[12 13] On the other hand, potentially low SES individuals had more difficulties obtaining a test even when they had sufficient health literacy. Our results also suggested that education level is a better determinant for COVID-19 status than income[29] because the association gradient was larger for low education participants than for low income participants. Further research is needed to fully understand the observed link between socioeconomic disparities and COVID-19.

Furthermore, there could be other explanations of the inverse association of SES and COVID-19 status. Despite the fact that the low SES group is at higher risk of noncommunicable disease, they also suffer from a higher general disease burden both mentally and financially during the pandemic.[26] Therefore, they might be less likely to notice the relevant symptom due to the already existing disease burden. Additionally, the first wave of COVID-19 pandemic

in the Netherlands covered the winter holiday, and people with high income and education were more likely to go for a relatively expensive skiing holiday compared with the low SES group. The skiing area in Italy has always been a popular destination for Dutch people and Northern Italy was the first outbreak centre during the first wave of this pandemic,[32] so that people with high income or high education who came back from the skiing holiday in Italy were more likely to self-report and be tested for COVID-19.

Interestingly, lifestyle factors only slightly mediated the pathway between SES and self-reported, and tested COVID-19 status, and did not seem to be a mediator in the pathway between SES and diagnosed COVID-19 status in our study, which is different from a previous study that suggested an unhealthy lifestyle that was a profound risk factor for COVID-19 status.[15] Some other studies also found that nutritional status[33 34] and smoking[2] might also be risk factors for COVID-19 disease severity, disease progression and mortality. Because of the differences of study population and COVID-19 outcome selection, our results were not comparable with previous studies. Nevertheless, this study provided a fresh message that in the general population, SES is a better determinant of self-reported, tested and diagnosed COVID-19

**Table 3** Association of income with different COVID-19 status

| Income | COVID-19 status | | | | | |
|---|---|---|---|---|---|---|
| | Self-reported | | Tested | | Diagnosed | |
| | OR (95% CI) | p | OR (95% CI) | p | RR (95% CI) | p |
| Model 1 | | | | | | |
| Low | 0.89 [0.83–0.96] | **0.002** | 0.85 [0.77–0.93] | **0.001** | 0.84 [0.54–1.31] | 0.4 |
| Middle | 0.82 [0.76–0.88] | **<0.001** | 0.84 [0.77–0.92] | **<0.001** | 1.14 [0.77–1.70] | 0.5 |
| High | Ref | | | | | |
| Model 2 | | | | | | |
| Low | 0.88 [0.81–0.95] | **0.001** | 0.87 [0.79–0.96] | **0.005** | 0.80 [0.51–1.25] | 0.3 |
| Middle | 0.82 [0.76–0.88] | **<0.001** | 0.86 [0.78–0.94] | **0.001** | 1.10 [0.73–1.64] | 0.6 |
| High | Ref | | | | | |
| Model 3 | | | | | | |
| Low | 0.88 [0.81–0.95] | **0.001** | 0.87 [0.79–0.96] | **0.006** | 0.80 [0.51–1.25] | 0.3 |
| Middle | 0.81 [0.76–0.87] | **<0.001** | 0.86 [0.78–0.94] | **0.001** | 1.10 [0.73–1.64] | 0.7 |
| High | Ref | | | | | |
| Model 4 | | | | | | |
| Low | 0.86 [0.79–0.93] | **<0.001** | 0.86 [0.78–0.95] | **0.002** | 0.74 [0.47–1.18] | 0.2 |
| Middle | 0.80 [0.74–0.87] | **<0.001** | 0.86 [0.78–0.94] | **0.002** | 1.05 [0.70–1.57] | 0.8 |
| High | Ref | | | | | |

Model 1: adjusted for income, age and sex. Model 2: adjusted for income, age, sex, and six lifestyle factors (smoking status, TV watching time ≥ 4h/day, sleep time <7 or >9 h/day, MVPA<150 min/week, LLDS, and heavy drinker). Model3: adjusted for income, age, sex, six lifestyle factors (smoking status, TV watching time ≥ 4h/day, sleep time <7 or >9 h/day, MVPA<150 min/week, LLDS, and heavy drinker), and BMI. Model 4: adjusted for income, age, sex, six lifestyle factors (smoking status, TV watching time ≥ 4h/day, sleep time <7 or >9 h/day, MVPA<150 min/week, LLDS, and heavy drinker), BMI, and ethnicity.
BMI, body mass index; LLDS, lifelines diet score; MVPA, non-occupational moderate-to-vigorous physical activity; OR, odds ratio; RR, risk ratio.

status. In addition, unlike other COVID-19 studies that found, obesity and ethnicity were both risk factors of COVID-19 infection, hospitalisation and mortality,[11 15 35–38] our results showed that obesity and ethnicity did not contribute substantially to the disparities in COVID-19 status, which indicates that obesity and ethnicity were not in the pathway between SES and self-reported, tested and diagnosed COVID-19 status.

This study has several strengths. First, this general population-based study about COVID-19 may to some extent mitigate the selection bias compared with studies conducted based in clinical settings and can allow us to thoroughly understand the relation between SES and COVID-19 in the general population. Second, we innovatively included a broader range of COVID-19 status, including self-reported and tested COVID-19 status, to better understand COVID-19-related health literacy and accessibility to health services. Still, this study failed to link the objective measures of COVID-19 infection status, which could cause some bias. Another limitation of the current study is the fact that the study population was predominantly white (more than 98%). Additionally, the Netherlands has a well-developed social security system. This may limit its generalisability to populations of other ethnicity, and in a different social context. One caveat of our study is that the relative low number of cases in the region, which is the nature of the COVID-19 outbreak in the Northern Netherlands (online supplemental table 4), might influence the statistical power of the study. However, we still find statistically significant results for SES and COVID-19 status. Another limitation is from the condensed observations about COVID-19 status since certain types of individuals might be more likely to fill in more questionnaires and are, therefore, more likely to report a positive COVID-19 status in at least one of the questionnaires. If these individuals predominantly belong to a specific SES group, this would bias the results. However, by further adjusting for the total number of questionnaires filled by each participants, we did not observe any difference from the current results (online supplemental table 1). Additionally, although we only observed minor effect of lifestyle factor on the association between SES and COVID-19 status, the true effect of lifestyle factors could be more pronounced since the low SES group tends to report their lifestyle according to the social desire. Finally, approximately 17% of the participants were dropped because of missing SES data, while 16% of the participants were dropped because of incomplete lifestyle information, which might bias the results given that people with low SES and worse lifestyle practices could be less likely to participant in the COVID-19 questionnaire. Nevertheless, both the study population and excluded participants have similar characteristics (online supplemental table 2). Consequently, it is less likely that there was a selection bias resulting from missing variables.

In conclusion, our findings of SES and self-reported, tested and diagnosed COVID-19 status indicate that individuals with lower SES may be more vulnerable and neglected during

COVID-19 pandemic, possibly because of disease burden, poor health literacy and access to healthcare, suggesting that health services and health promotion interventions should particularly focus on targeting individuals with lower SES to get them better prepared for future health emergencies.

**Author affiliations**
[1]Department of Internal Medicine, Division of Nephrology, University Medical Center Groningen, Groningen, The Netherlands
[2]Faculty of Economics and Business, University of Groningen, Groningen, The Netherlands
[3]Aletta Jacobs School of Public Health, University of Groningen, Groningen, The Netherlands
[4]Department of Epidemiology, University Medical Center Groningen, Groningen, The Netherlands
[5]Department of Genetics, University Medical Center Groningen, Groningen, The Netherlands
[6]Department of Medical Microbiology and Infection Prevention, University Medical Center Groningen, Groningen, The Netherlands

**Acknowledgements** The authors wish to acknowledge the services of the Lifelines cohort study, the contributing research centres delivering data to Lifelines, and all the study participants.

**Collaborators** Lifelines Corona Research initiative: Marike Boezen; Jochen Mierau; Lude Franke; Jackie Dekens; Patrick Deelen; Pauline Lanting; Judith Vonk; Ilja Nolte; Anil Ori; Annique Claringbould; Floranne Boulogne; Marjolein Dijkema; Henry Wiersma; Robert Warmerdam; Soesma Medema-Jankipersadsing.

**Contributors** YZ: study design, analysis design and execution, drafted the manuscript. MD: literature review, manuscript review and approval. HHD: study design, supervision, manuscript review and approval. RDF: study design, supervision, manuscript review and approval. LD: conception, supervision, manuscript review and approval. JM and Lifelines Corona Research initiative: conception, manuscript review and approval, stakeholder engagement.

**Funding** This project has received funding from the European Union's Horizon 2020 research and innovation programme under the Marie Skłodowska-Curie grant agreement No 754 425 as well as funding from NWO-Fast Track No 440.20.002. The Lifelines Biobank initiative has been made possible by funds from FES (Fonds Economische Structuurversterking), SNN (Samenwerkingsverband Noord Nederland) and REP (Ruimtelijk Economisch Programma).

**Competing interests** None declared.

**Patient consent for publication** Not required.

**Ethics approval** The Lifelines study is conducted according to the principles of the Declaration of Helsinki and approved by the Medical Ethics Committee of the University Medical Center Groningen, The Netherlands (2007/152). Before study entry, a signed informed consent form was obtained from each participant.

**Provenance and peer review** Not commissioned; externally peer reviewed.

**Data availability statement** Data may be obtained from a third party and are not publicly available. This study used two data sets: Lifelines Biobank and the Lifelines COVID-19 cohort. Researchers can apply for access to Lifelines data and/or samples through https://www.lifelines.nl/researcher/how-to-apply.

**ORCID iD**
Yinjie Zhu http://orcid.org/0000-0001-8059-6446

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
