## [Reviewer comments · BMJ Open]

ARTICLE DETAILS

TITLE (PROVISIONAL)	Association between socio-economic status and self-reported, tested, and diagnosed COVID-19 status during the first wave in the Northern Netherlands: A general population-based cohort from 49,474 adults
AUTHORS	Zhu, Yinjie; Duan, Mingjie; Dijk, Hermien; Freriks, Roel; Dekker, Louise; Mierau, Jochen

VERSION 1 – REVIEW

REVIEWER	Maria Grau Universitat de Barcelona, Spain
REVIEW RETURNED	11-Jan-2021

GENERAL COMMENTS	The aim of this manuscript is to investigate the associations of SES with self-reported, tested, and diagnosed COVID-19 status in the general population. The manuscript has several flaws that the authors should address. Published studies should appear in Introduction, instead of Medrxiv studies that has not been peer reviewed. Please include the recent most evidence of the association between SES and COVID-19 in your Introduction (e.g. Baena-Diez JM. J Public Health (Oxf). 2020). Please take care with your statements: "Still, an in-depth investigation of the lifestyle and socio-economic 82 determinants of COVID-19 in the general population is lacking because current COVID-19 studies mostly relied on individuals presenting with symptoms in a clinical setting and focused on post COVID-19 infection status (e.g., hospitalization and intensive care)". The study performed by Baena-Diez et al. did not have these characteristics... Part of Introduction (information about questionnaires) should be placed in Methods. In addition, "As such, the findings of this study are less likely to be biased because of the population selection in clinical settings and can allow us to thoroughly understand the relation between SES and COVID-19 in the general population" should be placed in Strengths and Limitations. Was the COVID-19 questionnaire validated? The authors stated that this is a less biased methodology. However, this statement could be fake if the questionnaire is not validated. Please, be more cautious in your statements "This paper is the first to provide insight in socio-economic disparities on self-reported COVID-19, tested COVID-19, and diagnosed COVID-19 in the general population". Please delete this sentence. You're not sure that you're the first and it means nothing, in fact.
--

REVIEWER	Scott Montgomery Örebro University, Sweden
REVIEW RETURNED	25-Jan-2021

GENERAL COMMENTS	Zhu and colleagues investigated if markers of socioeconomic position and lifestyle markers are associated with self-reported measures of SARS-CoV-2 infection and testing. I am uncertain about some of the conclusions, which might benefit from elaboration.  1. The inverse association of measures of education and income with the infection outcomes could have a number of explanations, such as a genuine inverse association (in some countries the first to be infected had returned from relatively expensive skiing holidays in Italy), a higher general disease burden among the most disadvantaged, so symptoms are less likely to be noticed, or poor health literacy and access to some aspect of healthcare. This should be clarified, the limitation of no objective measure of infection stressed and the conclusion nuanced. 2. The authors assume that the inverse associations of education and income with the infection outcomes signals lack of awareness of infection, thus there is underreporting of the outcomes among the most disadvantaged. If this is so, the investigation of how lifestyle factors contribute to adverse outcomes is problematic and a major limitation. We might expect to see greater risk of infection among those with poorer lifestyle aspects, but this will be hidden in these data. This should be explained, and the results reported extremely cautiously. 3. I wonder about the theoretical model, as indicated by the statistical analysis section. Lifestyle factors are described as potential confounding factors for the association of socioeconomic position with the infection outcomes. Socioeconomic position itself does not influence such outcomes directly but operates through a range of living conditions and behaviour. I think the measures described as potential confounding factors are possible mechanisms linking/mediating the associations. This should be clarified, and the analysis altered accordingly. 4. Another sentence on the Lifelines study should be added to the methods section to provide a little more information on the structure of the cohort, such as birth cohorts/years, generations etc, so readers do not have to search for a reference for a basic understanding of the study. 5. I suspect there is typing error in this sentence on page 3. This study might fail to identify the ethnic differences because the study population was predominantly white (more than 98%).
---

VERSION 1 – AUTHOR RESPONSE

Detailed point-by-point response to the reviewers' comments

Reviewer: 1

Dr. Maria Grau, Institut Hospital del Mar d'Investigacions Mediques Comments to the Author: The aim of this manuscript is to investigate the associations of SES with self-reported, tested, and diagnosed

COVID-19 status in the general population. The manuscript has several flaws that the authors should address.

Point 1: Published studies should appear in Introduction, instead of Medrxiv studies that has not been peer reviewed. Please include the recent most evidence of the association between SES and COVID-19 in your Introduction (e.g. Baena-Diez JM. J Public Health (Oxf). 2020).

Response 1: Thank you for pointing it out, we agree with the reviewer that it is not proper to include Medrxiv studies that has not been peer reviewed in our introduction. To accommodate the comment of the reviewer, we have removed the two studies of Abedi Vida *et al.* and Wiemers, E.E. *et al.* from Medrxiv, and included the most recent evidence of the association between SES and COVID-19 in our introduction, i.e., Baena-Diez JM. J Public Health (Oxf). 2020 and Lieberman-Cribbin, W., et al., American Journal of Preventive Medicine, 2020 (line 83). Accordingly, we have removed the reference from Medrxiv in the discussion part and replaced with peer-reviewed most recent evidence.

Point 2: Please take care with your statements: "Still, an in-depth investigation of the lifestyle and socio-economic determinants of COVID-19 in the general population is lacking because current COVID-19 studies mostly relied on individuals presenting with symptoms in a clinical setting and focused on post COVID-19 infection status (e.g., hospitalization and intensive care)". The study performed by Baena-Diez et al. did not have these characteristics.

Response 2: We thank the reviewer for reminding us to take care with the above statement. Indeed, we have neglected some literature that have also focused on a general population, e.g, Baena-Diez et al. Therefore, to accommodate this comment, we have revised the sentence in line 89-91 and the revised text reads as follows "Still, an in-depth investigation of the lifestyle and socio-economic determinants of COVID-19 in the general population is needed because current COVID-19 studies mainly relied on individuals presenting with symptoms in a clinical setting and focused on post COVID-19 infection status (e.g., hospitalization and intensive care)". Accordingly, we also rephrased the sentence in line 92.

Point 3: Part of Introduction (information about questionnaires) should be placed in Methods. In addition, "As such, the findings of this study are less likely to be biased because of the population selection in clinical settings and can allow us to thoroughly understand the relation between SES and COVID-19 in the general population" should be placed in Strengths and Limitations.

Response 3: We agree with the reviewer that the information about questionnaires in introduction should be placed in methods. We therefore have moved this part to line 106-110 in methods section and adapted the content where appropriate. We also agree that "As such, the findings of this study are less likely to be biased because of the population selection in clinical settings and can allow us to thoroughly understand the relation between SES and COVID-19 in the general population" should be placed in Strengths and Limitation, so we have relocated and modified this part to line 292-249 in Strengths and Limitation section.

Point 4: Was the COVID-19 questionnaire validated? The authors stated that this is a less biased methodology. However, this statement could be fake if the questionnaire is not validated.

Response 4: We thank the reviewer for this question regarding the validation of the COVID-19 questionnaire. After consulting the Lifelines management group and checking relevant resources, we understand that the COVID-19 questionnaire covers many research areas, and some parts of the questionnaires include validated questionnaires. The outcomes of this study were derived from three subjective questions from the questionnaire, but we have cross checked with the national and regional data (supplementary table 1), which can be seen as a validation.

Point 5: Please, be more cautious in your statements "This paper is the first to provide insight in socio-economic disparities on self-reported COVID-19, tested COVID-19, and diagnosed COVID-19 in the general population". Please delete this sentence. You're not sure that you're the first and it means nothing, in fact.

Response 5: Thanks for pointing this out, the reviewer is correct, indeed we should be more cautious in our statements. To accommodate the comment, we have deleted the sentence and replaced with

“This paper has provided evidence and insight for socio-economic disparities in self-reported COVID-19, tested COVID-19, and diagnosed COVID-19 in a general population.” in line 231-234.

Reviewer: 2

Prof. Scott Montgomery, Örebro University Comments to the Author:
Zhu and colleagues investigated if markers of socioeconomic position and lifestyle markers are associated with self-reported measures of SARS-CoV-2 infection and testing. I am uncertain about some of the conclusions, which might benefit from elaboration.

Point 1. The inverse association of measures of education and income with the infection outcomes could have a number of explanations, such as a genuine inverse association (in some countries the first to be infected had returned from relatively expensive skiing holidays in Italy), a higher general disease burden among the most disadvantaged, so symptoms are less likely to be noticed, or poor health literacy and access to some aspect of healthcare. This should be clarified, the limitation of no objective measure of infection stressed and the conclusion nuanced.

Response 1: We thank the reviewer for pointing this out and we agree with the reviewer that the explanation of the inverse association of education and income with the infection outcomes should be more clarified. Indeed, we should mention the possible explanation about the skiing holidays in Italy and about the general disease burden among the most disadvantaged population. To accommodate this comment, we have clarified more about the explanation in the discussion section (line 265-275). Moreover, we also agree that we lack the objective measures of the infection status, thus we have included this limitation in our discussion section (line 296-297), and revised our conclusion accordingly (line 46-48 and line 318-320).

Point 2. The authors assume that the inverse associations of education and income with the infection outcomes signals lack of awareness of infection, thus there is underreporting of the outcomes among the most disadvantaged. If this is so, the investigation of how lifestyle factors contribute to adverse outcomes is problematic and a major limitation. We might expect to see greater risk of infection among those with poorer lifestyle aspects, but this will be hidden in these data. This should be explained, and the results reported extremely cautiously.

Response 2: Thanks for raising this concern. Indeed we think that there is underreporting of the outcomes among the most disadvantaged because of lack of awareness of infection, or the already existing disease burden. In addition, the most disadvantaged people also tend to report their lifestyle according to the social desire, thus their lifestyle factors may be overreported. As mentioned by the reviewer, this fact that lifestyle factor did not really influence the association between SES and COVID-19 status might be hidden in the overreported lifestyle factors even though we still observe a minor attenuation of the coefficient after adjustment for lifestyle factors, still, we have mentioned this in the limitation part (line 309-312) to accommodate the reviewer's comment.

Point 3. I wonder about the theoretical model, as indicated by the statistical analysis section. Lifestyle factors are described as potential confounding factors for the association of socioeconomic position with the infection outcomes. Socioeconomic position itself does not influence such outcomes directly but operates through a range of living conditions and behaviour. I think the measures described as potential confounding factors are possible mechanisms linking/mediating the associations. This should be clarified, and the analysis altered accordingly.

Response 3: We agree with the reviewer that lifestyle factors are possible mechanisms linking/mediating the association, that is why we adjusted the lifestyle factors in model2 to see the attenuation of the coefficient, the percentage of attenuation kind reflect the linking effect of the lifestyle factors, which is also the basis of mediation analysis. To accommodate the reviewer's comment, we have tested the mediation analysis with lifestyle factors together as the mediator. The proportional mediated was shown in the following table if the mediation effect was significant. Additionally, we have added this table in the supplementary file (Supplementary Table 3) and adjusted the description of statistical analysis in the method section (line 189-196), results (line 218-220, 225-227), and discussion (line 249-250) accordingly.

SES indicators	COVID-19 Status		
	Self-reported	Tested	Diagnosed
Education			
Proportion mediated (%)*	9.9	ns	ns
Income			
Proportion mediated (%)	6.7	9.5	ns

As shown in the table, lifestyle factors only slightly mediated the association of SES and self-reported, and tested COVID-19 status because the proportion mediated was really small (<10%). This was in accordance with our regression model² that the coefficient was slightly attenuated after adjustment for lifestyle factors for self-reported and tested COVID-19 status.

Point 4. Another sentence on the Lifelines study should be added to the methods section to provide a little more information on the structure of the cohort, such as birth cohorts/years, generations etc, so readers do not have to search for a reference for a basic understanding of the study.

Response 4: We thank the reviewer for pointing this out, we agree with the reviewer that another sentence on the Lifelines study should be added to the methods section to provide a little more information on the structure of the cohort. To accommodate this comment, we have added extra information about the recruitment and structure of the Lifelines cohort study in line 115-123.

Point 5. I suspect there is a typing error in this sentence on page 3. This study might fail to identify the ethnic differences because the study population was predominantly white (more than 98%).

Response 5: Thanks for pointing this out, we have revised the typing error in the sentence in line 54, now the sentence reads as follows "This study might fail to identify the ethnic differences because the study population was predominantly white (more than 98%)".

VERSION 2 – REVIEW

REVIEWER	María Grau University of Barcelona, Spain
REVIEW RETURNED	15-Feb-2021
GENERAL COMMENTS	I have no further comments
REVIEWER	Scott Montgomery Örebro University, Sweden.
REVIEW RETURNED	02-Mar-2021
GENERAL COMMENTS	My comments have been addressed by the authors.